# Mitochondria-Mediated Programmed Cell Death in *Saccharomyces Cerevisiae* Induced by Betulinic Acid is Accelerated by the Deletion of *PEP4* Gene

**DOI:** 10.3390/microorganisms7110538

**Published:** 2019-11-07

**Authors:** Hongyun Lu, Qin Shu, Hanghang Lou, Qihe Chen

**Affiliations:** Department of Food Science and Nutrition, Zhejiang University, Hangzhou 310058, China; luhongyun@zju.edu.cn (H.L.); louhanghang@zju.edu.cn (H.L.); 21713041@zju.edu.cn (Q.S.)

**Keywords:** *Saccharomyces cerevisiae*, Pep4p, betulinic acid, mitochondria-mediated programmed cell death

## Abstract

In this work, using *Saccharomyces cerevisiae* as a model, we showed that BetA could inhibit cell proliferation and lead to lethal cytotoxicity accompanying programmed cell death (PCD). Interestingly, it was found that vacuolar protease Pep4p played a pivotal role in BetA-induced *S. cerevisiae* PCD. The presence of Pep4p reduced the damage of BetA-induced cells. This work implied that BetA may induce cell death of *S. cerevisiae* through mitochondria-mediated PCD, and the deletion of *Pep4* gene possibly accelerated the effect of PCD. The present investigation provided the preliminary research for the complicated mechanism of BetA-induced cell PCD regulated by vacular protease Pep4p and lay the foundation for understanding of the Pep4p protein in an animal model.

## 1. Introduction

Currently, yeast is a well-established unicellular eukaryotic model organism used to elucidate the molecular mechanism of apoptotic pathways and has offered new insights into human diseases. Betulinic acid (BetA), a plant derived pentacyclic lupine-type triterpene, possesses a variety of biological activities, including inhibition of human immunodeficiency virus (HIV) [1], antibacterial [2], antimalarial [3], anti-inflammatory [4], and anthelmintic properties [5]. Moreover, the remarkable selectivity cytotoxicity induced by BetA in a diversity of cancer cells together with its minimal cytotoxicity for normal cells has raised this compound as an optimistic candidate for a non-toxic anti-cancer drug [6]. Previous investigations have shown that in mammal cells, the death receptor pathway is not involved in BetA-induced cell death [7]. In addition, BetA-induced cell death is independent of p53 [6,7]. Repeated evidence supports the inference that BetA induces cell death via the mitochondrial pathway [8,9], regulated by the Bcl-2 family proteins [10]. BetA appears to directly target the mitochondrial PT pore, inducing cytochrome c release [11]. Besides, the disruption of the mitochondrial membrane potential and production of reactive oxygen species (ROS) have been reported to be associated with the presence of BetA [12].

Apoptosis is a genetically regulated self-destruction program essential for both development and tissue homeostasis in multicellular organisms (the major form of PCD). In yeast, a mitochondria-mediated PCD pathway shows similarities to the mammalian intrinsic pathway. Consistently, mitochondrial outer membrane permeabilization (MOMP) is as decisive an event in the PCD process as in mammalian cells due to the release of pro-apoptotic factors like cytochrome C [13], a PCD inducing factor (AIF) [14], and Endo G [15]. Mitochondria are the main place where ROS is generated. Generally speaking, ROS generation was induced under high membrane potential [16]. MOMP is related to the redox state of mitochondria and could also be mediated by ROS [17,18]. *Saccharomyces cerevisiae* proteinase A (PrA) is a member of the aspartic proteinase superfamily. During the nutritional stress, sporulation, and vegetative growth conditions, PrA is found to be essential to the *S. cerevisiae* vacuolar proteolytic system [19,20]. In the previous research, Pep4p was found to be a regulating factor for cell physiology and metabolic processes in *S. cerevisiae* under a nitrogen stress environment [21]. As is well-acknowledged, yeast vacuoles, membrane-bound acidic organelles, which share many similarities to plant vacuoles, and mammalian lysosomes are involved in the regulation of PCD process. The vacuolar protease Pep4p (also called yeast proteinase A), orthologous of the human CatD [22], contributes to yeast programmed cell death [23,24,25]. Pep4p is an aspartic protease found in the yeast vacuole, which is important for protein turnover after oxidative damage. Under the stress of H_2_O_2_, the overexpression of Pep4p prevents the accumulation of oxidized proteins without increasing lifespan. In contrast, protein degradation and the removal of oxidized proteins were decreased in Pep4p-deficient cells [26]. Pep4p migrates out of vacuoles without major vacuolar rupture and nucleoporin degradation [27]. Pep4p translocates to the cytosol with both vacuolar and plasma membrane integrities, which is also involved in mitochondrial degradation during acetic acid-induced PCD [28]. These results suggest that Pep4p has a role in apoptotic cell death and the release of a vacuolar protease during regulating cell death is conserved in yeast. In addition, Pereira et al. [29] found that the protective role of Pep4p in acetic acid-induced PCD depended on its catalytic function associated with the presence of AAC proteins. However, the molecular role of Pep4p in mitochondrial degradation and its involvement in the process of PCD process remains understood. In this work, we used BetA as the major compound to induce the cell PCD process, firstly showing that vacuolar Pep4p can provide protection for *S. cerevisiae* cells PCD via a mitochondria-mediated pathway.

## 2. Materials and Methods

Betulinic acid (>97% pure, Tokyo Chemical Industry Co., Ltd., Tokyo, Japan) was dissolved at 4 mg/mL in DMSO (Sigma-Aldrich, St Louis, MO, USA) and aliquots were stored at −80 °C.

### 2.1. Yeast Strains and Growth Conditions

Several *Saccharomyces cerevisiae* strains were investigated in this study, including BY4741 (MATa, his3Δ1, leu2Δ0, met15Δ0, ura3Δ0) and BY4741 pep4Δ (the mutant containing the knock-out of pep4 gene encoding proteinase A). The two strains were grown in YPD medium (1% yeast extract, 2% peptone, 2% glucose) at 28 °C to a logarithmic phase (about 4 × 10^6^ CFU/mL). Logarithmic phase cells were harvested and washed in phosphate-buffered saline (PBS) (pH 7.4 ± 0.2) buffer for three times, then suspended in PBS containing 20 μg/mL DMSO (solvent blank) or 20 μg/mL BetA, following an incubation time of 2 h, 4 h, and 8 h. The yeast culture cells were harvested for further experiments.

### 2.2. Growth and Survival Tests

Cell growth was monitored by cell spots assay. The collected cells were washed three times with 1 × PBS buffer, adjusted to identical optical density (OD_600_), and diluted 10^−1^, 10^−2^, 10^−3^, 10^−4^, 10^−5^, respectively. Then, 5 μL of each diluted yeast culture cells were spotted onto YPD plates. Plates were incubated for 2 d and the number of grown colonies was calculated. Cell viability was tested by Cell Blue Cell Viability Assay Kit (Huabio, Hangzhou, China). This kit performs as well as other resazurin-based cell proliferation assay kits like AlamarBlue reagent [30]. In brief, 1 × 10^7^ cells were seeded onto 96-multiwell plates together with 10 μg/mL BetA and incubated for different times at 28 °C. 10 μL cell blue solutions was then added and placed in the same incubator for 4 h. Subsequently, cell growth was assayed by the absorbance using fluorescence followed in a fluorescence microplate reader (SpectraMax M5, Molecular Devices, Sunnyvale, CA, USA). The excitation wavelength was 560 nm and the emission wavelength was 590 nm.

### 2.3. 4,6-Diamidino-2-phenylndole (DAPI) Staining and Microscopy

The DAPI staining of nuclei tissue was adopted as described previously [28]. Yeast cells were harvested and washed three times with 1×PBS buffer, and re-suspended in 70% ethanol for brief fixation and permeabilization. After incubation at 4 °C for 2 h, cells were washed three times with 1× PBS buffer and incubated with 1 μg/mL of DAPI in the dark at room temperature for 10 min, and then rinsed three times with PBS. The stained cells were mounted on a coverslip and observed with an inverted Zeiss laser scanning confocal microscope (Zeiss LSM-780, Carl Zeiss Inc., Thornwood, NJ, USA) equipped with a 63× oil immersion lens.

### 2.4. ROS Detection

ROS production was quantified by flow cytometry using dihydroethidium (DHE) (Beyotime, China) staining. Cells were collected by centrifugation, washed two times, re-suspended in 500 μL PBS buffer and incubated with 10 μg/mL DHE for 30 min in the dark. Cells with red fluorescence were analyzed using an inverted laser scanning confocal microscope (Zeiss LSM-780, Carl Zeiss Inc.) or a FACSCalibur (BD Biosciences, San Jose, CA, USA) with excitation at 535 nm and emission at 610 nm.

### 2.5. Mitochondrial Transmembrane Potential (ΔΨmt) Assay

The mitochondrial transmembrane potential (ΔΨ) was evaluated in cells treated with the laser dye rhodamine 123 (Beyotime, Nanjing, China). Cultured cells were washed twice using PBS and treated with rhodamine 123 at a final concentration of 10 µg/mL. The cells were incubated at 37 °C for 30 min in the dark. After that, the cultured cells were washed at least 3 times with PBS buffer before FACS analysis using a BD FACSCalibur (BD Biosciences, San Jose, CA, USA) equipped with an FL-1 channel (488 nm/530 nm).

### 2.6. Annexin V/PI Staining

Phosphatidylserine exposure was detected by an Annexin V/PI apoptosis kit (MultiSciences, Shanghai, China), as described by Madeo et al [31] with minor modifications. Briefly, 1–5 × 10^6^ cells were collected by centrifugation, and washed in sorbitol buffer (1.2 M sorbitol, 0.5 mM MgCl_2_, 35 mM potassium phosphate, pH 6.8), digested with 50 U/mL lyticase (Sigma Chemical Co., St Louis, MO, USA) [32] in sorbitol buffer for 2 h at 28 °C, harvested, washed in binding buffer (10 mM Hepes/NaOH, pH 7.4, 140 mM NaCl, 2.5 mM CaCl_2_) containing 1.2 M sorbitol buffer, harvested, and resuspended in binding buffer/sorbitol. Afterwards, 5 µL annexin-FITC and 10 µL propidium iodide (PI) were added to 500 µL cell suspension, and then were incubated for 5 min at room temperature. Lastly, the cells were measured with an inverted laser scanning confocal microscope (Zeiss LSM-780, Carl Zeiss Inc., Dublin, CA, USA) or a FACSCalibur (BD Biosciences).

### 2.7. Terminal Deoxynucleotidyl Transferased UTP Nick-End Labeling (TUNEL)

Yeast cells were fixed with 3.7% formaldehyde, digested by lyticase, and applied to a polylysine-coated slide as described for immunofluorescence analysis. The slides were rinsed with PBS, incubated in permeabilization solution (0.1% Triton X-100, 0.1% sodium citrate) for 2 min on ice, rinsed twice with PBS, incubated with 50 µL TUNEL reaction mixture (50 µL TdT, 450 µL fluorescein-dUTP, Boehringer Mannheim Roche) for 60 min at 37 °C, rinsed three times with PBS buffer, incubated with 50 mL converter-POD for 30 min at 37 °C, rinsed three times with PBS, and then the TUNEL reaction was visualized by chromogenic staining with DAB (Sigma, St. Louis, MO, USA). Slides were counterstained with hematoxylin.

### 2.8. Transmission Electron Microscopy

TEM samples were prepared as described previously with some modifications [33]. Cells were harvested by gentle centrifugation, washed in phosphate-buffered saline (PBS) (pH 7.2), resuspended in 2.5% (*v*/*v*) glutaraldehyde and fixed for 2 d at 4 °C. Cells were further fixed in a 2% solution of osmium tetroxide in 0.1 mM phosphate buffer. Fixed cells were dehydrated with 30%, 50%, 75%, 85%, 95%, and 100% ethanol, gradually. After the 100% ethanol washes, cells were washed using 100% acetone, infiltrated with 50% acetone/50% Epon for 1 h, then with 100% Epon for 3 h. Cells were transferred to fresh Epon (100%) and incubated at 56 °C for 48 h. Ultrathin sections were prepared and observed under a TEM (JEM-1230, JEOL, Tokyo, Japan) after staining with lead acetate. Micrographs were recorded using a JEM-1230 electron microscope (JEOL Ltd, Tokyo, Japan) at a magnification of 8000.

### 2.9. Real Time Quantitative PCR (qPCR)

Total RNA was extracted with Trizol reagent (Invitrogen, Carlsbad, CA, USA), referred to the manufacturer’s instructions. The quality and concentration of the extracted RNA were assessed by spectrophotometer and electrophoresis in 1% agarose. DNase digestion and cDNA synthesis use PrimeScript RT Reagent Kit with gDNA Eraser (Takara, Dalian, China). cDNA was synthesized from 1 μg total RNA. qPCR was performed with the synthesized cDNA as the template. All primer sequences were listed in Table A1. The efficiencies and specificities of the primers were tested by dilution experiments and melting curves, respectively. qPCR examinations were performed using SybrGreen qPCR Master Mix (Rainbio, Shanghai, China). Reactions were conducted using an ABI7500 Fast Real-Time PCR System (Applied Biosystems, Foster City, CA). The parameters for PCR were composed of pre-incubation at 95 °C for 2 min; 40 cycles of amplification steps at 95 °C for 10 s and 60 °C for 40 s; and then a cooling step at 50 °C for 30 s. At the end of the amplification cycle, a melting analysis was conducted to verify the specificity of this reaction. The experiments were performed in biological replicates with an independent measurement of each sample and mean values were used for further calculations. The fold changes were determined by the 2−^ΔΔ*C*T^ method normalized to the *ACT1* gene [34].

## 3. Results

### 3.1. Vacuolar Pep4p Provides the Protective Effect on BetA-induced Cell Death of S. Cerevisiae

Firstly, to examine the effect of BetA on yeast cells viability, the results showed that with the treating time of BetA extending, the viability of two strains decreased (Figure 1A). After exposure to BetA for 4 h, the differences of cell viability between BetA-treatment groups and the control groups were significant (Figure 1B). After 12 h, the differences disappeared, thus we chose 2, 4, and 8 h for further experiments. Interestingly, the Δpep4 groups shared similar viability with the wild-type ones at each time point, but the survival rate of Δpep4 groups was much lower than that of the wild-type groups before 12 h (Figure 1C). To assess the effect of BetA on cell growth, the cells pre-treated with 20 μg/mL BetA for different times were incubated in YPD. The aliquots of the cells were harvested at indicated time points for CFU counting. The result demonstrated that BetA induced cell death in a time-dependent manner (Figure 1D). Moreover, we found that the Δpep4 mutant was much more sensitive to BetA treatment (Figure 1D). These data suggest that BetA could reduce cell viability and inhibit cell growth. Moreover, the hypersensitivity of Δpep4 cells to BetA may be due to lacking the protection of vacuolar Pep4p in yeast cells.

### 3.2. Pep4p Reduces the Hallmarks of PCD Process Induced by BetA

Chromatin condensation and fragmentation were usually considered as a typical marker of the apoptosis process. To determine whether the BetA-induced cell death was consistent with PCD, we examined the morphology change of the chromatin after treatment by BetA. Commonly, fragments were aligned as a ring close to the nuclear envelope [35]. The BetA-treated wild-type strain as well as the Δpep4 one was stained with DAPI. The result showed that DNA fragments arranged as a half-ring at the inner side of nuclear envelope (Figure 2B,E,F), and several randomly distributed nuclear fragments of the cells (Figure 2B,C,E,F). Δpep4 cells showed more significant chromatin condensation coincident with nucleus fragmentation. The nuclei of the untreated cells were homogeneous in cell shape and cell density (Figure 2A,D).

Meanwhile, Annexin V/propidium iodide (PI) co-staining was used to quantify externalization of phosphatidylserine, an early apoptotic event (Annexin V positive), and membrane permeabilization, which is indicative of necrotic death (Annexin V negative, PI positive). Cells exposed to BetA were frequently stained green at the cell periphery after digestion of cell wall, whereas a few cells in more advanced apoptotic or necrotic stages stained green and red due to the inability of cell membrane to exclude PI (Figure 3A). Flow cytometry results showed that about 13.24% of the Δpep4 cells were Annexin V positive or both Annexin V and propidium iodide positive after the treatment by BetA, but only 7.15% showed the same result for wild-type cells. DMSO groups revealed fewer apoptotic cells in contrast to the BetA-treated groups, respectively (Figure 3B). Meantime, these four groups all showed a low necrotic rate (Figure 3B).

Apoptotic stimuli have been reported to cause cleavage of chromosomal DNA [31]. Therefore, we checked whether BetA-induced cells death was associated with DNA cleavage. It was observed that after treatment by BetA for 4 h, approximately 50% Δpep4 cells formed a strong TUNEL-positive phenotype (Figure 4D). 30% wild-type cells also revealed TUNEL-positive phenotype (Figure 4B). The DMSO group of wide-type cells showed no TUNEL-positive phenotype (Figure 4A), meanwhile Δpep4 cells treated with DMSO presented few TUNEL-positive phenotype (Figure 4C). This means that DMSO showed little damage on yeast cells at the designed concentration. Carmona-Gutiérrez et al. found that the CatD ortholog (Pep4p) of the budding yeast S. cerevisiae harbors a dual cytoprotective function, composed of an anti-apoptotic part and an anti-necrotic part [25]. Based on the above data, it is clear that vacuolar Pep4p in *S. cerevisiae* reduces the hallmarks of apoptosis induced by BetA.

### 3.3. Pep4p Shows Significant Protective Effect on Cell Ultrastructure Upon BetA Treatment

TEM has been used to uncover definitive ultrastructural features of the treated cells. *S. cerevisiae* cells exposure to BetA demonstrated various morphological characteristics of PCD. The nuclei of DMSO group cells showed normal cellular morphology with a distinct cell wall, an intact nucleus with clear double membrane structure (Figure 5A,a). In contrast, electron microscopic images of cells incubated with 20 μg/mL BetA revealed extensive chromatin condensation (Figure 5B,b), multiple nuclear fragments (Figure 5B), and irregular structures protruding from the nucleus (Figure 5B,C,c). Other abnormal structures embedded with intact membrane in the vacuoles (Figure 5C,D,b). But these abnormal structures were not observed in DMSO-treated cells. In addition, some cells possessed tiny vesicles on the outer side of the plasma membrane (Figure 5B,D,E,b–d). Other cells exhibited the distended cell wall and ruptured internal organelles (Figure 5E,d). The organelles and cytoplasm were clearly degraded (Figure 5F,d), cytoplasmic membrane disappeared and cell plasma exuded (Figure 5F). Further, Δpep4 cells treated by BetA were damaged more seriously. Some cells appeared cavities inside of the vacuole (Figure 5c,e). Cell walls disintegrated, entocyte exuded and formed a large void (Figure 5f).

### 3.4. PCD Induced by BetA was Accompanied by Perturbation of Mitochondria in the Presence of Pep4p

ROS is a leading regulator in mitochondria-dependent PCD of yeast cell, which is considered as a necessary step in the development of apoptotic cascade [36,37]. In this work, we evaluated ROS formation after pre-treated with BetA using both laser scanning confocal microscope and flow cytometry analysis. The results demonstrated that intracellular levels of ROS promoted with the increase of incubating time (Figure 6B). Both the wild-type and the Δpep4 cells treated with BetA showed an obvious increase in the amount of ROS, which was much higher than that of DMSO groups, respectively (Figure 6B). Besides, the Δpep4 cells incubated with BetA revealed significant ROS production, in contrast to that of the BetA-treated wild-type cells (Figure 6A,B). The data suggest that BetA may stimulate the production of intracellular ROS, whereas pep4p may resist its accumulation.

As stated earlier, mitochondrion plays an important role in yeast PCD process [38]. BetA induced a transient mitochondrial membrane potential hyperpolarization followed by a depolarization. An early event of PCD generally attributed to the loss of mitochondrial membrane potential (ΔΨm), for it represents the opening of the mitochondrial permeability transition pore and release of apoptogenic factors. To assess the levels of ΔΨm in the cells exposed to BetA, cells were collected after incubation for different treatment times and then stained with Rh123. We found that after incubation for 2 h, ΔΨm presented a low level. While after incubation for 4 h, ΔΨm level increased rapidly. Subsequently, the ΔΨm level revealed a decline trend (Figure 7). What is more, BetA-treated groups (both the WT and the Δpep4 ones) revealed much lower mitochondrial membrane potential if compared to DMSO-treated groups. Though shown with low ΔΨm, the cells remained the specific mitochondria stain after treated by BetA for 8 h, which indicating that the mitochondria membrane integrity was still preserved.

### 3.5. Significance of BetA on the Expression of Mitochondria Related Genes

Mitochondria were an important organelle not only for respiration but for PCD also. It was reported that in mammal cells, BetA induced PCD via triggering mitochondria directly, subsequently activating caspases [7,39,40]. To investigate the effect of BetA on two strains, qPCR was used to elucidate the underlying mechanism of yeast PCD induced by BetA. In this work, six genes, *Aif1, Cyc1, Cyc7, Ndi1, Dnm1, and Fis1* involved in mitochondrial PCD were checked. As stated in Figure 8, in the wild-type cells treated with BetA, *Cyc7* and *Ndi1* genes were remarkably up-regulated, whereas *Aif1* and *Cyc1* genes were down-regulated when compared to the cells in DMSO (Figure 8A). Interestingly, in Δpep4 cells group treated by BetA, most genes except for *Aif1* were up-regulated in comparison to the DMSO group (Figure 8B). Moreover, the comparison between the Δpep4 cells treated by BetA and the wild-type cells treated by BetA was conducted, we found that all investigated genes were up-regulated in Δpep4 cells after the treatment of BetA (Figure 8C). Relevant data demonstrated that both mitochondrial AAC (ADP/ATP carrier) proteins and vacuolar Pep4p interfere with mitochondrial degradation [28], which suggests a complex interplay between mitochondria and the vacuole in yeast programmed cell death.

## 4. Discussions

In this study, we have shown that BetA caused cytotoxicity, and Pep4p protein provided the protective role in resisting BetA treatment by *S. cerevisiae* cells. The cytotoxicity of BetA is marked by typical apoptotic phenotypes, including chromatin condensation, DNA fragmentation and externalization of phosphatidylserine. Though both the wild-type cells and the Δpep4 ones treated with BetA showed apoptotic phenotypes, the extent of the PCD that developed was different. BetA caused chromatin condensation, tiny vesicles on the outer side of the plasma membrane, ruptured internal organelles, and disappeared cytoplasmic membrane in the wild-type cells. However, besides the features pointed out above, in Δpep4 cells we found cavities inside of the vacuole and disintegrated cell walls, which were unrecoverable for the cells. This may explain the reason why the survival rate of Δpep4 cells was much lower than that of wild-type ones. The postulated interacting mode of BetA and Pep4p protein against yeast cells PCD were presented in Figure 9. It is well acknowledged that BetA is an interesting natural compound that is capable of killing plenty of tumor cells [41,42]. It is clear that Pep4p in vaculoe certainly affected the resistibility of yeast cells to stress factors.

In the meantime, we have demonstrated that BetA also induced perturbation of mitochondria, including alteration of mitochondrial membrane potential, the generation of ROS, and expression of mitochondria related genes. In mammal cells, an increase in mitochondrial membrane potential is followed by several types of PCD [43,44,45]. In yeast, such phenomena were shown in the PCD to be induced by pheromone and amiodarone [46]. An elevation of ΔΨm increases ROS production that initiated the mitochondrial thread-grain transition and de-energization. The mitochondrial de-energization finally results in loss of ΔΨm, accompanied by the release of PCD factors from mitochondria to cytosol. Meanwhile, high concentrations of ROS can cause detrimental oxidative stress through inflicting oxidative damage on essential biomolecules such as nucleic acids, proteins, and lipids [47] that play a key role in the PCD process [36,48,49]. In this work, we found that BetA induced a transient mitochondrial membrane potential hyperpolarization, followed by a depolarization in the two strains. Mason et al [27] reported that Pep4p can migrate out of vacuoles under the shock of H_2_O_2_ and Δpep4 cells are not protected from H_2_O_2_-induced cell death. Pep4p protected yeast cells from H_2_O_2_ damage generally through removing oxidized proteins [26]. Herein, under the same treatment condition, intracellular ROS level in Δpep4 cells was much higher than that of the wild-type cells. Though it was hard to identify whether Pep4p was involved in oxidized proteins degradation directly or catalyze other proteinase like Sod1p or Sod2p hydrolyze the damaged proteins, our data combined with previous studies implied that Pep4p was essential for yeast cells in response to several stresses.

As well, mitochondrial proteins have critical roles in the process of yeast PCD process. These proteins are components of the electron transport chain or of the inner or outer mitochondrial membranes (IMM and OMM, respectively). Herein, six kinds of mitochondrial proteins were assessed, including Ndi1p, Cyt c, Aif1p, Dnm1p, and Fis1p. Yeast Ndi1, a ubiquinone oxidoreductase, displays the most sequence homology (28% identity overall) to AMID. Ndi1 is involved in PCD induced by various stimuli investigated, including H_2_O_2_, Mn, and acetate acid. Ndi1 catalyzes the intramitochondrial NADH and produce ROS [50]. While the disruption of Ndi1 is involved in PCD induced by various stimuli investigated, including H_2_O_2_, Mn, and acetate acid. Ndi1 resulted in an increased life span, the overexpression of Ndi1 led to the appearance of typical apoptotic markers [51]. Cui et al [52] indicated that when cells under environmental stress- or drug-induced, Ndi1 is cleaved at its N-terminal to be activated in mitochondria, and the truncated Ndi1 is released to the cytoplasm to provoke PCD. In this study, Ndi1 was remarkably up-regulated both in wide-type and in Δpep4 cells after treatment by BetA, which was in agreement with the observation of ΔΨm hyperpolarization and ROS generation. Yeast cytochrome c has two isoforms, iso-1 and iso-2-cytochrome c, encoded by the nuclear *cyc1* and *cyc7* genes, respectively [53]. Release of Cyt c from the mitochondria in response to pro-apoptotic stimuli was observed in both mammal and yeast cells [11,13,47,54,55,56]. Once released from mammalian mitochondria, it forms a complex in the cytosol with the PCD-protease activating factor 1 to form a structure that is called the apoptosome [57]. The apoptosome binds up to seven caspase-9 proteins activating them. BetA-induced mammalian cells PCD shares the same change. However, in yeast, the lethal role of Cyt c is uncertain and the relation between Cyt c release and caspase activation requires further investigation [55,58]. Meanwhile, we found that whether *Pep4* gene was absent, BetA could induce *Cyc7* up-regulation in comparison to DMSO groups. As compared to wild-type cells, Cyc genes were up-regulated in Δpep4 cells. In addition, the expression of *Dnm1* and *Fis1*, which was the factors controlling mitochondrial fission [59], was influenced. In wild-type cells, the fold changes of both genes were slight. Obviously, the mitochondrial dynamics were much extensive in Δpep4 cells. What makes us interesting was that after the treatment by BetA, the expression of *Aif1* gene was down-regulated compared to DMSO groups. Multiple stimulators could induce *Aif1* release from mitochondria, like acetic acid, H_2_O_2_, and chronological lifespan [14]. Moreover, the apoptotic function of Aif1p was partially caspase dependent [14]. Though qPCR data showed the involvement of mitochondrial proteins in yeast cells and that PCD is correlated with *Pep4* gene in response to BetA, more evidence still should be obtained through future research. Pereira et al. [25] found that the vacuolar protease Pep4p played a role in mitochondrial degradation using yeast genetic approaches. Depletion and overexpression of *Pep4* gene delays and enhances mitochondrial degradation, respectively. Moreover, Pep4p is released from the vacuole into the cytosol in response to acetic acid treatment. Besides, the results reported by Marques et al. also interestingly demonstrated that Pep4p was important for protein turnover after oxidative damage; However, the increased removal of oxidized proteins is not sufficient to enhance cell lifespan [26]. In summary, we demonstrated that BetA could induce yeast PCD in a manner of ways depending on the mitochondria. What is more, the presence of vacuolar Pep4p protein could reduce the damage caused by BetA. It is indicated that *Pep4* gene interacts with the PCD process induced by BetA against yeast cells. The findings in this work showed a potential to regulate cancer or tumor cells PCD by modifying orthologous of the human CatD proteins. Future study should focus on the mammanlian model investigation using CatD protein to verify the present conclusions.

## Figures and Tables

**Figure 1 microorganisms-07-00538-f001:**
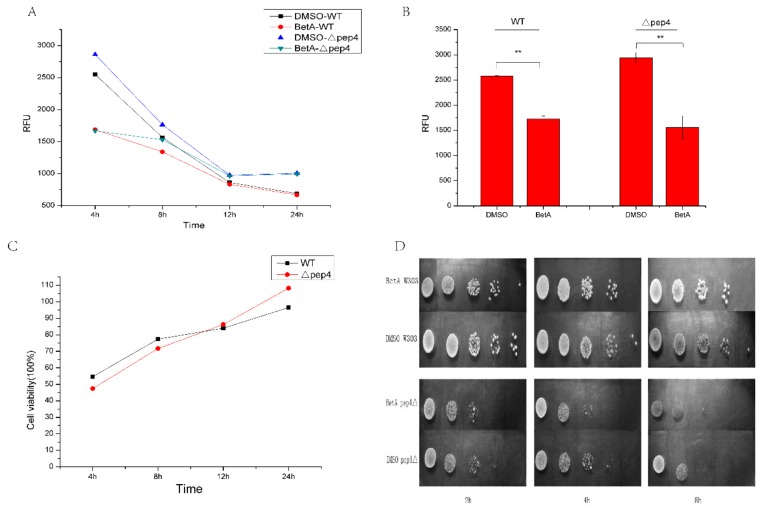
Effect of 20 μg/mL BetA on cell viability and survival of two yeast strains. Data = mean ± SD from three independent experiments, ** *p* < 0.01. (**A**) Proliferation activity expressed in relative fluorescence units (RFU) after 1 × 10^7^ cells were exposed to 20 μg/mL BetA for different incubation times. After exposure to 20 μg/mL BetA for different times, 10 μL Cell Blue solutions were added into the medium. Then cells were incubated at 37 °C for 4 h. Measure absorbance using fluorescence with excitation wavelength at 560 nm and emission wavelength at 590 nm using a fluorescence plant reader. (**B**) Proliferation activity expressed in relative fluorescence units (RFU) after exposure to 20 μg/mL BetA for 4 h. (**C**) The cell viability of Δpep4 groups and wild-type groups. (**D**) Spot dilution assays in YPD medium (2 μL per spot). Yeast cells were spotted on corresponding positions of different plates by serial 5-fold dilutions. The plates were incubated at 28 °C for 48 h.

**Figure 2 microorganisms-07-00538-f002:**
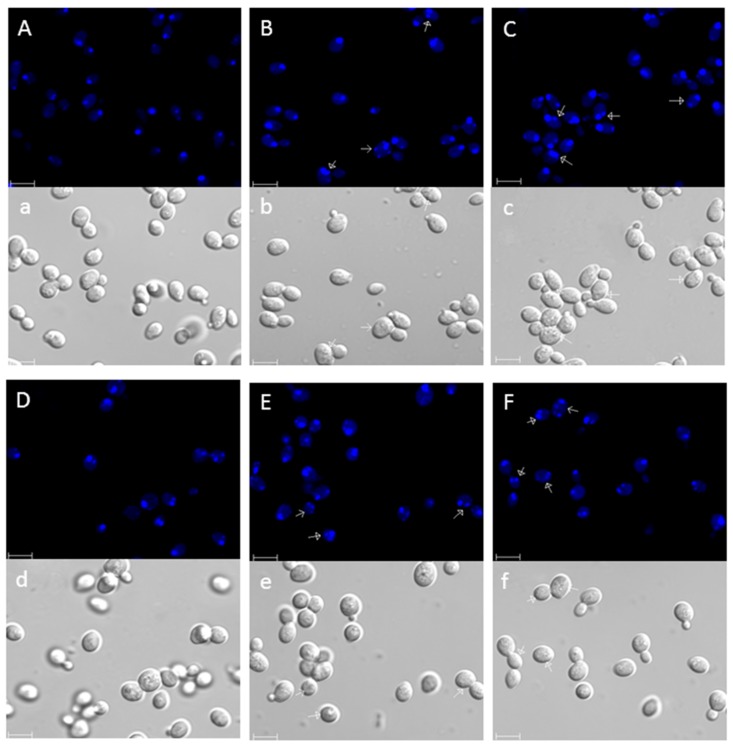
DAPI-stained S. cerevisiae nuclei. (**A**–**C**) W303 cells, (**D**–**F**) Δpep4 cells. (**A**,**D**) control group; (**B**,**C**,**E**,**F**) cells treated with 20 μg/mL of BetA for 4 h. Arrows showed half-ring nuclear fragments and closed arrows showed randomly distributed nuclear fragments. Bar, (**a**–**f**,**A**–**F**), 5 μm.

**Figure 3 microorganisms-07-00538-f003:**
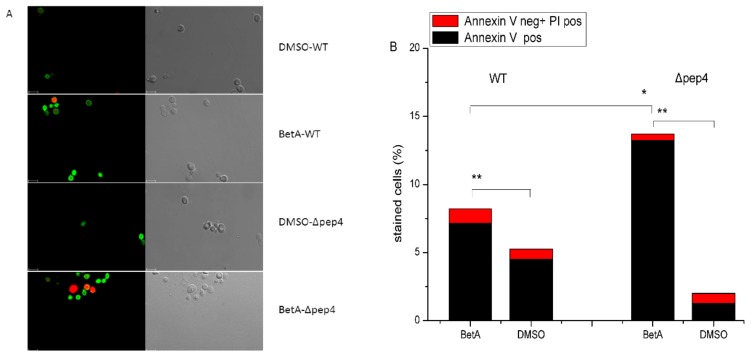
Annexin V/PI costaining of S. cerevisiae. Phosphatidylserine externalization and membrane integrity were detected 4 h after incubation with 20 μg/mL BetA. Stained cells were quantified by laser scanning confocal microscope (**A**) or flow cytometry (**B**), the rates of stained cell by Annexin V were 7.15%, 4.58%, 13.24%, and 1.33%, respectively, in the four samples. In each experiment, at least 10,000 cells were evaluated. Bar, 5 μm. * *p* < 0.05, ** *p* < 0.01.

**Figure 4 microorganisms-07-00538-f004:**
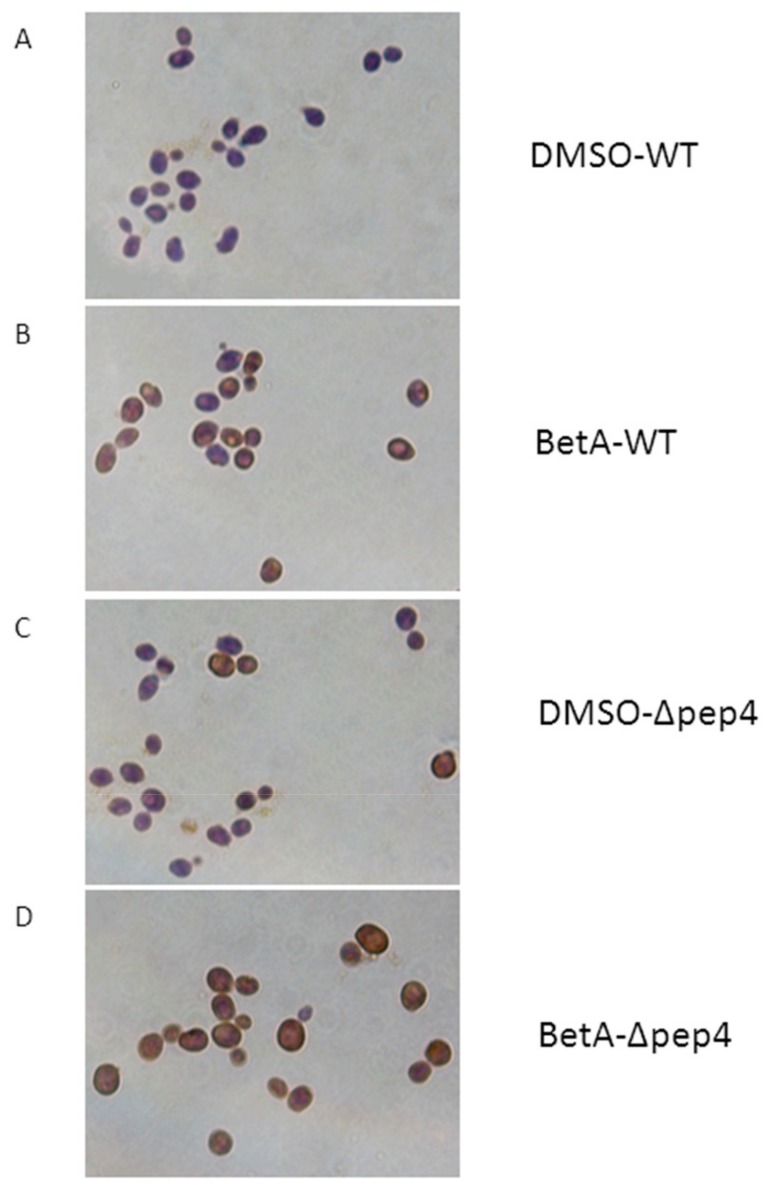
TUNEL-stained *S. cerevisiae* nuclei. 1 × 10^7^ cells were incubated with 20 µg/mL BetA for 4 h. Thereafter, yeast cells were stained with TUNEL. TUNEL-positive cells showed brown, while TUNEL-negative cells showed blue, magnification, 1000×. (**A**) The TUNEL -stained DMSO-WT group, (**B**) The TUNEL -stained BetA-WT group, (**C**) The TUNEL-stained DMSO-Δpep4 group, (**D**) The TUNEL -stained BetA- Δpep4 group.

**Figure 5 microorganisms-07-00538-f005:**
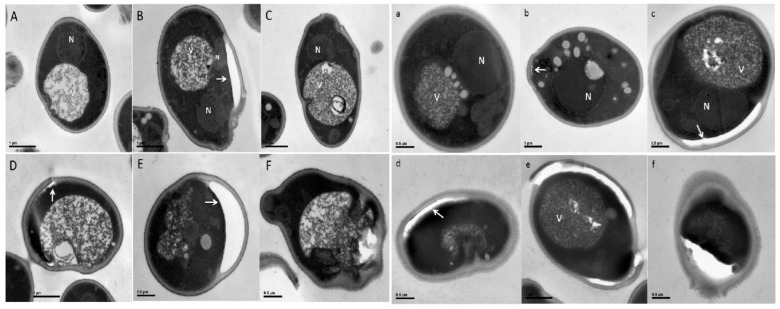
Electron microscopy images of *S. cerevisiae*. Upper graphs were the wild-type yeast groups; Lower graphs were the Δpep4 yeast groups. (**A**,**a**), the control group; (**B**–**F**), wild-type cells; (**b**–**f**), Δpep4 cells. Cells were treated with 20 µg/mL of BetA for 4 h. Bar, (**A**–**D**,**b**,**e**), 1 µm; (**e**,**f**,**a**,**c**,**d**,**f**,**E**,**F**), 0.5 µm. Arrows show the vesicles on the outer side of the plasma membrane.

**Figure 6 microorganisms-07-00538-f006:**
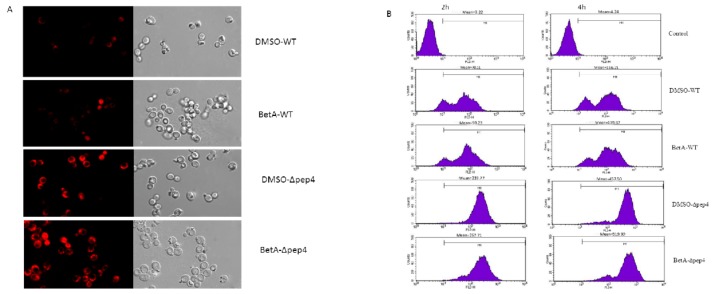
BetA induced generation and accumulation of ROS for diferent yeast strains. (**A**) 1 × 10^7^ cells were incubated with 20 µg/mL BetA for 4 h subsequently analyzed for ROS accumulation by DHE staining with laser scanning confocal microscope. Bar, 5 mm. (**B**) 1 × 10^7^ cells were incubated with 20 µg/mL BetA for 2 h or 4 h, subsequently analyzed for ROS accumulation by flow cytometry.

**Figure 7 microorganisms-07-00538-f007:**
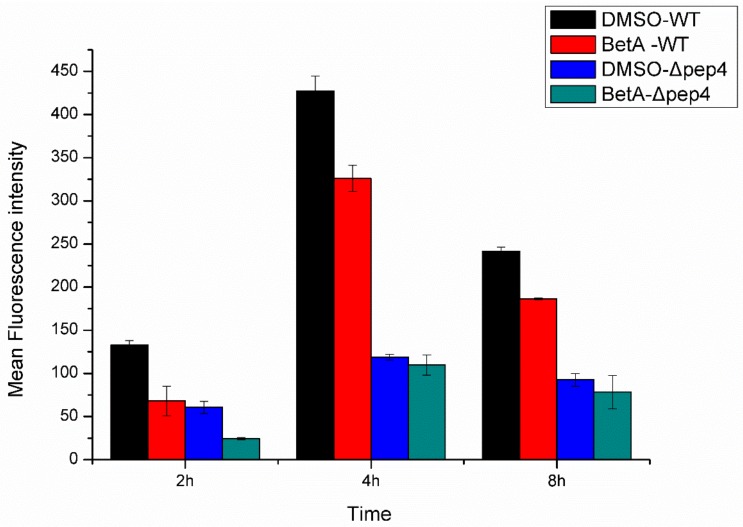
The change of S. cerevisiae mitochondrial membrane potential for four experimental groups. 1 × 10^7^ CFU/mL cells were incubated with 20 µg/mL BetA for 2 h, 4 h, and 8 h, respectively. Thereafter, yeast cells were stained with Rhodamine-123 (10 µM) for 30 min and then analyzed in FL-1 channels of flow cytometer.

**Figure 8 microorganisms-07-00538-f008:**
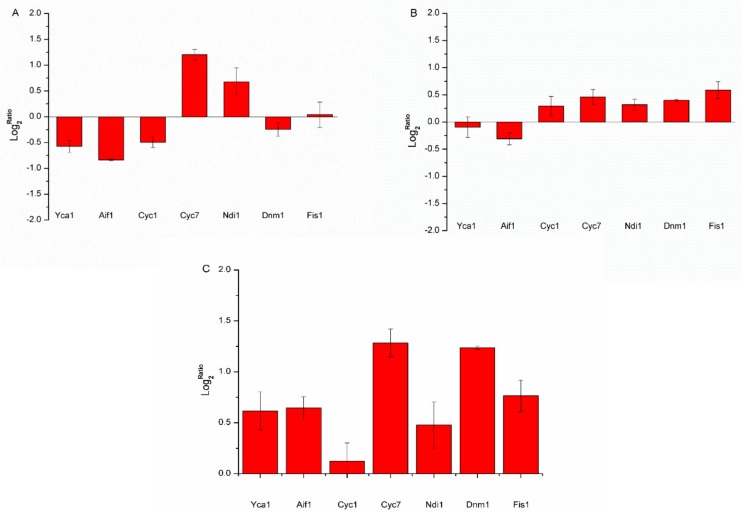
Fold changes (Log_2_) in key genes involved in mitochondrial PCD, measured by qPCR. The cells were treated using 20 μg/mL BetA for 4 h. The treatment by 20 μg/mL DMSO was used as the solvent blank. RNAs were purified from the samples collected and used in qPCR analyses. Data are presented as Log2Ratio and are normalized to actin gene. If the Log_2_Ratio=0, the expression level of this gene was not changed. (**A**) Compared with the DMSO group, seven genes changes after BetA treatment in the wild-type strain. (**B**) Compared with the DMSO group, seven genes changes after BetA treatment in the Δpep4 strain. (**C**) Compared with the wild-type strain treated by BetA, seven genes changes after BetA treatment in Δpep4 one. Each value represents the mean of three independent measurements.

**Figure 9 microorganisms-07-00538-f009:**
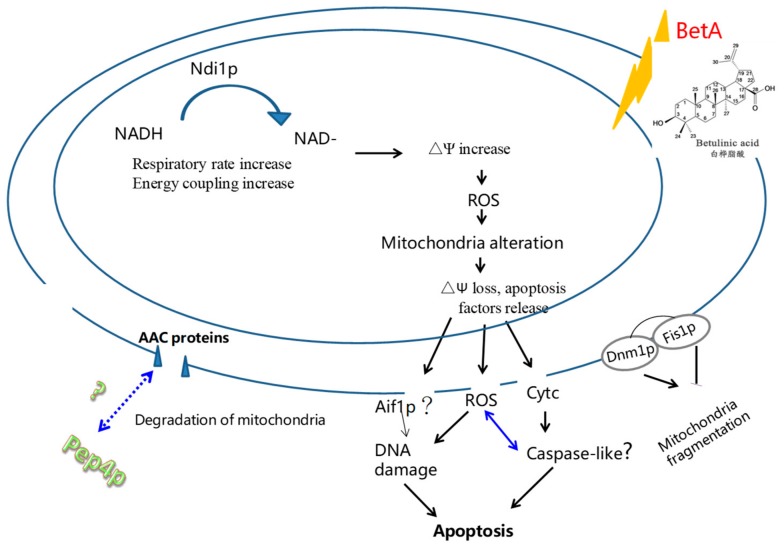
The postulated reaction pathway by BetA-induced PCD in yeast cells through a mitochondria mediated pathway. What is more, the Pep4p provide yeast cells a protection for its cleaning function. 
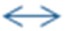
 represent a defined interaction, 
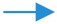
 represent a progressive flow relation, 
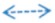
 represents the interaction to be determined.

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
