# Peer review of "Mitochondria-Mediated Programmed Cell Death in Saccharomyces Cerevisiae Induced by Betulinic Acid is Accelerated by the Deletion of PEP4 Gene"

_microorganisms, 2019, doi:10.3390/microorganisms7110538_

Round 1

Reviewer 1 Report

The authors have done a good job to characterize the cell death induced by betulinic acid and the deletion of PEP4 gene. However, I cannot support publication of this manuscript at this point, because the authors interpret this cell death as apoptosis. In contrast to their interpretation the analyzed cell death program has no resemblance to apoptotic cell death in mammalian cells. Therefore, calling this cell death program apoptosis seems disingenuous. If the authors, however, choose to change ‘apoptosis’ to ‘programmed cell death’ or ‘cell death program’ throughout the manuscript, I believe the manuscript should be considered for publication.

Author Response

If the authors, however, choose to change ‘apoptosis’ to ‘programmed cell death’ or ‘cell death program’ throughout the manuscript, I believe the manuscript should be considered for publication.

Response: Thank you for your constructive and helpful suggestion. The relevant subtitles have been revised in the revision.

Reviewer 2 Report

The manuscript by Lu et al. entitled “Mitochondria-mediated apoptosis in Saccharomyces cerevisiae induced by betulinic acid is accelerated by the deletion of PEP4 gene” describes studies showing apoptotic characteristics in yeast upon betulinic acid (BetA) treatment and how the presence of yeast protease A (Pep4p), through the use of a pep4 gene knockout, reduces the resulting effects of BetA.  The authors have done an excellent job in providing a clear research objective for their study involving yeast in relation to other current mammalian studies, and provide a foundation for future studies involving the role of similar proteases in other cell types.  A few minor comments are described below.

No reference to figure 1C is given the results section (3.1) when referring to figure 1. The authors should consider adding the Annexin+/PI+ data to figure 3B. This would be informative to the reader as the authors refer to this in the text. The description of figure 7 in the text (lines 258-269) is unclear. It appears that the delta-pep4 strain is more depolarized at any given time compared to the WT strain.  Additionally, the extent of further depolarization is limited in the delta-pep4 strain at 4 and 8 hours.  This draws into question the effect BetA has on the mitochondrial membrane potential in the absence of Pep4p).  Is the data shown in figure 8C normalized to the controls? Grammar the sentence structure should be checked.

Author Response

No reference to figure 1C is given the results section (3.1) when referring to figure 1.

Response: Thank you for your constructive and helpful suggestion. The sentence:“Interestingly, the Δpep4 groups shared similar vialibility with the wild-type ones at each time point, but the survival rate of Δpep4 groups was much lower than that of the wild-type groups before 12h (Fig. C).”has been marked in line167-169 on page 6.

The authors should consider adding the Annexin+/PI+ data to figure 3B. This would be informative to the reader as the authors refer to this in the text.

Response: Thank you for your constructive and helpful suggestion. “the rates of stained cell by Annexin V were 7.15%, 4.58%, 13.24%, and 1.33%, respectively, in the four samples” has been added in line 215-216 on page 7.

The description of figure 7 in the text (lines 258-269) is unclear. It appears that the delta-pep4 strain is more depolarized at any given time compared to the WT strain. Additionally, the extent of further depolarization is limited in the delta-pep4 strain at 4 and 8 hours. This draws into question the effect BetA has on the mitochondrial membrane potential in the absence of Pep4p)

Response: Thank you for your constructive and helpful suggestion. In the absence of Pep4p group, compared to the DMSO-treated group, the BetA treatment group showed lower mitochondrial membrane potential levels

Is the data shown in figure 8C normalized to the controls?

Response: Thank you for your constructive and helpful suggestion. Yes, the data shown in Fig. 8C has been normalized to the controls.

Grammar the sentence structure should be checked.

Response: Thank you for your constructive and helpful suggestion.  The relevant subtitles have been revised in the revision.

Round 2

Reviewer 1 Report

My critizism remains unaddressed.

Author Response

If the authors, however, choose to change ‘apoptosis’ to ‘programmed cell death’ or ‘cell death program’ throughout the manuscript, I believe the manuscript should be considered for publication.

Response: Thank you for your constructive and helpful suggestion. I'm sorry that we didn't thoroughly realize what you meant before and made some incorrect modifications.

Firstly, we were faced with the question: Is it really apoptosis that is observed in yeast cells when compared to mammalian cell death? If apoptosis is defined as autonomous cell death in connection with certain morphological and cytological markers, yeast certainly performs apoptosis. If the definition demands the involvement of specific proteins such as caspases or bcl-2 relatives, yeast does not. However, the similar response of yeast and animals to the expression of several pro- and anti-apoptotic genes argues strongly in favour of a common origin of the processes. Yeast has already shown its value as a model for apoptosis research [1]. Moreover, similar anti-apoptotic genes have been found in yeast and mammals [2], an anti-apoptotic function of the mammalian CDC48 orthologue VCP has been described, confirming the effects of the cdc48S565G mutation in yeast as bona fide apoptosis [3], this is the earliest report on yeast cell apoptosis. Afterwards, yeast has been successfully used as a model system to elucidate several aspects of apoptosis regulation in mammalian cells, particularly of the intrinsic mitochondrial pathway [4]

In this study, our results have showed that with the increase of the treatment time of BetA, the cell viability of yeast cells decreased gradually, and the mortality increased with the outcomes of apoptosis characterization, including formation of nuclear debris, chromatin contraction, phosphatidic acid serine valgus, and breakage. Given phosphatidylserine valgus (Annexin V positive) is a hallmark of early cell apoptosis, moreover, the annexin V fluorescein isothiocyanate/propidium iodide (PI) kit, DNA fragmentation, as well as loss of mitochondrial membrane potential analysis used in our study, is often applied to detect the apoptosis of human cell or yeast as well[5-8]. Therefore, we judged that the death of yeast cells was apoptosis in morphologically.

In order to make the research more scientific and rigorous, we also did apoptotic bodies research. However, we did not detect apoptotic bodies produced by apoptosis, which may be engulfed by neighboring cells. Therefore, it’s also doubtful whether we use apoptotic or programmed death to describe yeast cell death. Programmed cell death (PCD) referring to an active process of cell death to maintain the stability of internal environment after receiving some signal or being stimulated by some factors, includes apoptotic, autophagy, necroptosis and pyroptosis. In our study, the treatment of BetA could induce cell death of yeast, while the control group (without BetA treatment) still maintained good cell activity under the same treatment conditions. In addition, the presence of pep4 in yeast cells can protect yeast cells exposed to BetA treatment and slow down the damage of BetA to yeast cells. Therefore, at this level, we considered your suggestion is reasonable that change apoptosis to PCD.

[1] K.-U. Fröhlich, F. Madeo, Apoptosis in yeast – a monocellular organism exhibits altruistic behaviour, FEBS Letters, 473 (2000) 6-9.

[2] D. Williams, G. Norman, C. Khoury, N. Metcalfe, J. Briard, A. Laporte, S. Sheibani, L. Portt, C.A. Mandato, M.T. Greenwood, Evidence for a second messenger function of dUTP during Bax mediated apoptosis of yeast and mammalian cells, Biochimica et Biophysica Acta (BBA) - Molecular Cell Research, 1813 (2011) 315-321.

[3] T. Shirogane, T. Fukada, J.M.M. Muller, D.T. Shima, M. Hibi, T. Hirano, Synergistic roles for Pim-1 and c-Myc in STAT3-mediated cell cycle progression and antiapoptosis, Immunity, 11 (1999) 709-719.

[4] M.J. Sousa, F. Azevedo, A. Pedras, C. Marques, O.P. Coutinho, A. Preto, H. Gerós, S.R. Chaves, M. Côrte-Real, Vacuole-mitochondrial cross-talk during apoptosis in yeast: A model for understanding lysosome-mitochondria-mediated apoptosis in mammals, Biochemical Society Transactions, 39 (2011) 1533-1537.

[5] J. Shang, L. Wu, Y. Yang, Y. Li, Z. Liu, Y. Huang, Overexpression of Schizosaccharomyces pombe tRNA 3′-end processing enzyme Trz2 leads to an increased cellular iron level and apoptotic cell death, Fungal Genetics and Biology, 122 (2019) 11-20.

[6] Y.-S. Kim, H.-O. Jin, S.-K. Seo, S.H. Woo, T.-B. Choe, S. An, S.-I. Hong, S.-J. Lee, K.-H. Lee, I.-C. Park, Sorafenib induces apoptotic cell death in human non-small cell lung cancer cells by down-regulating mammalian target of rapamycin (mTOR)-dependent survivin expression, Biochemical Pharmacology, 82 (2011) 216-226.

[7] V.M. Martins, T.R. Fernandes, D. Lopes, C.B. Afonso, M.R.M. Domingues, M. Côrte-Real, M.J. Sousa, Contacts in Death: The Role of the ER–Mitochondria Axis in Acetic Acid-Induced Apoptosis in Yeast, Journal of Molecular Biology, 431 (2019) 273-288.

[8] M. Jovanovic-Tucovic, L. Harhaji-Trajkovic, M. Dulovic, G. Tovilovic-Kovacevic, N. Zogovic, M. Jeremic, M. Mandic, V. Kostic, V. Trajkovic, I. Markovic, AMP-activated protein kinase inhibits MPP+-induced oxidative stress and apoptotic death of SH-SY5Y cells through sequential stimulation of Akt and autophagy, European Journal of Pharmacology, 863 (2019) 172677.